# Prey Capturing Dynamics and Nanomechanically Graded Cutting Apparatus of Dragonfly Nymph

**DOI:** 10.3390/ma14030559

**Published:** 2021-01-25

**Authors:** Lakshminath Kundanati, Prashant Das, Nicola M. Pugno

**Affiliations:** 1Laboratory of Bio-Inspired, Bionic, Nano, Meta Materials and Mechanics, Department of Civil, Environmental and Mechanical Engineering, University of Trento, Via Mesiano 77, 38123 Trento, Italy; l.kundanati@unitn.it; 2Mechanical Engineering Department, University of Alberta, 116 St. and 85 Ave., Edmonton, AB T6G 2R3, Canada; pdas@ualberta.ca; 3School of Engineering and Materials Science, Queen Mary University of London, Mile End Road, London E1 4NS, UK

**Keywords:** dragonfly nymph, labium, mandible, nanoindentation, high-speed videography

## Abstract

Aquatic predatory insects, like the nymphs of a dragonfly, use rapid movements to catch their prey and it presents challenges in terms of movements due to drag forces. Dragonfly nymphs are known to be voracious predators with structures and movements that are yet to be fully understood. Thus, we examine two main mouthparts of the dragonfly nymph (Libellulidae: Insecta: Odonata) that are used in prey capturing and cutting the prey. To observe and analyze the preying mechanism under water, we used high-speed photography and, electron microscopy. The morphological details suggest that the prey-capturing labium is a complex grasping mechanism with additional sensory organs that serve some functionality. The time taken for the protraction and retraction of labium during prey capture was estimated to be 187 ± 54 ms, suggesting that these nymphs have a rapid prey mechanism. The Young’s modulus and hardness of the mandibles were estimated to be 9.1 ± 1.9 GPa and 0.85 ± 0.13 GPa, respectively. Such mechanical properties of the mandibles make them hard tools that can cut into the exoskeleton of the prey and also resistant to wear. Thus, studying such mechanisms with their sensory capabilities provides a unique opportunity to design and develop bioinspired underwater deployable mechanisms.

## 1. Introduction

Aquatic insects are one of the most diverse groups of animals in small water bodies. In this group of insects, there is a huge diversity in the broad classification of predators and prey, and some of the top predators have been identified as diving beetles, bugs and dragonfly larvae [1] Aquatic predation presents a complex scenario of mechanics, primarily because of the hydrodynamic drag forces experienced during the motion of the predator and its prey-capturing mechanism. Some of these predatory insects use rapid movements to catch the prey. Such rapid movements in many animals are expected to be an outcome of evolutionary pressures such as escaping from predators and catching fast prey [2]. An example of such a kind of movements is the predatory strike of a mantis shrimp [3]. Dragonfly nymphs are another example, which are voracious predators that feed on tadpoles, mosquito larvae and other smaller aquatic organisms [4]. They are proven to be a potential bio-control agent of mosquitoes [5]. They are also the only insects or animals that are known to have a modified underlip that is used as a protractile organ, referred to as a labium.

Prey-capturing by a dragonfly nymph involves visually locking onto the prey to be captured and, in certain cases, pressure build-up in the abdomen with water that helps generate quick forward movement using propulsion. Earlier studies suggest that prey detection may also be achieved by sensing the underwater currents with the use of mechanoreceptors on the legs of the nymphs of *L.depressa* [6]. The predatory strike in dragonfly nymphs is performed by using the labium that is modified into a prehensile mask for capturing the fast-moving prey [7]. This involves a gradual opening up of the two cup-shaped labial palpi and moving them forward with the help of unfolding prementum and postmentum parts, similar to the extension of human arms with the elbow and shoulder joints. Depending on the distance of the prey from the mouthpart, the forward movement was observed to be assisted by the movement of the body. Once the palpi reach around the prey, they are brought together to form a closed cup shape to grab the prey. After the prey is captured, it is drawn towards the mouth to be fed on. The process of capturing the prey involves two primary mechanisms. Firstly, the self-propulsion that is achieved by generating jets from the anal valve, and secondly, the labium protraction for capturing the prey. Both the mechanisms are hypothesized to function based on similar hydraulic dynamics [8]. These hydraulic mechanisms are generated by the use of several muscles on both sides of the abdominal diaphragm, thereby generating pressure waves as required. The mechanics of the labial extension of Anax Imperator (*Aeshnidae*) were earlier analyzed [9], and the inertial forces and torques acting on the labium were estimated. Additionally, the labium protraction was hypothesized to be initiated based on a click mechanism at the prementum–postmentum joint that disengages to release energy [10]. In terms of the mechanism energetics, the labium protraction occurs by releasing the energy stored, using strong contraction of the abdomen and thorax muscles [11]. The shape of the labial palpi of the globe skimmer larvae (*Pantala flavescens*) used in the present study is different from the shape of the earlier studied species. This feature has been reported in previous studies [4]; however, the real-time observation and sensory structures of such a mechanism have not been reported yet.

The mouthparts of dragonfly nymphs include the extendable labium, mandibles, labrum, maxillae and hypopharynx [12]. In this study, we focus on the labium and mandibles, which serve important functions such as capturing the prey and cutting it. The other mouthparts are known to serve in manipulating and tasting the food. In the present work, we examined the rapid movement of the labium of globe skimmers (*Pantala flavescens*) nymphs that aid in prey-capturing. We first recorded the nymphs under water using high-speed photography to observe the process of labium protraction and retraction in detail and later estimate the time scales involved. We also characterized the morphological details of the labium to examine the shape and the sensory structures. Additionally, we measured the mechanical properties of the mandibles of the nymph that are used in cutting the captured prey. In underwater manipulators, control is very difficult because of the various hydrodynamic effects influencing underwater [13]. Our study helps to shed further light on the biomechanical aspects of the dragonfly nymph predation and thereby possible development of bioinspired underwater deployable manipulators.

## 2. Experimental Materials and Methods

### 2.1. Specimen Collection and Videography

The nymphs of globe skimmers (Insecta: Odonata: Libellulidae) were collected using nets from local, natural and artificial ponds. These nymphs were kept in containers with water collected from the corresponding ponds. The nymphs from the penultimate instar stage were used in the experiments. The nymphs were then transferred to an acrylic tank of dimensions 25 cm (length) × 10 cm (width) × 10 cm (depth) for high-speed video recording of the predation process. We used mosquito larvae collected from similar ponds to use as bait for the predator nymphs. Single bait was then laid at some distance from the nymph, and the entire process of predation thereafter was recorded using a 1 megapixel high-speed camera (Phantom Miro 110).

### 2.2. Microscopy

The images of nymph mouthparts were taken using an optical microscope (Lynx LM-1322, OLYMPUS, Tokyo, Japan). Images were captured using a CCD camera (Nikon) attached to the microscope. The dimensions from the images were reported using a standard scale bar in a microscope that was checked with calibration.

SEM imaging was performed directly on the samples, which were stored in 100% alcohol and then air dried. Prepared labia were carefully mounted on double-sided carbon tape, stuck on an aluminum stub and sputter coated (Manual sputter coater, Agar scientific) with gold. An SEM (EVO 40 XVP, Zeiss, Oberkochen, Germany) was used with accelerating voltages between 5 and 20 kV. ImageJ software was used for all dimensional quantifications reported in this study.

### 2.3. Microindentation and Nanoindentation

Mandible samples separated from the nymphs were embedded in a resin and polished using a series of 400-, 800-, 1200-, 2000- and 4000-grade sandpapers. Finally, the sample was polished using a diamond paste of particle sizes in the range of 6 µm and 1 µm, to obtain a surface of minimal roughness. Poisson ratio of 0.31 was used to estimate the elastic modulus [14].

Microindentation experiments (N = 2 mandibles from different nymphs) were performed using a standard CSM micro indenter (Vickers) with a load application 50 mN and at a loading rate of 100 mN/min. We used two mandibles from two different nymphs for the microindentation experiments. In nanoindentation experiments (N = 2 mandibles from different nymphs), a Berkovich indenter was used to perform indentations up to a maximum load of 5 mN at the rate of 1800 mN/min. NanoBlitz3D software was used to map the cross-sectional surface of the tooth samples.

## 3. Results and Discussion

### 3.1. Microstructure of the Mouthparts

The prey-capturing labium constitutes a postmentum (*pmt*), prementum (*prmt*) and labial palpi (*Plp*), which are connected to the prementum by articulation joints (Figure 1A,B). These parts are moved in co-ordination using concentrated action of the muscles, the articulation joints and membranous structures [7]. A snapshot taken from the high-speed video shows the protraction of the labium with the change in angle of the prementum and postmentum of the labium, and the opening of labial palpi for capturing the prey (Figure 1C,D). We also observe that both the prementum and postmentum are supported by cuticular tendon-like structures running through them (Figure 1C,D). Such tendon-like elements are hypothesized to facilitate in the storage and release of elastic energy in rapid movement musculatures of insects [15].

The extendable mouthpart labium primary includes prementum, postmentum and palpi (Figure 2A). The palpi are two halves (L and R in Figure 2A) that can open and close by moving in a vertical plane using the corresponding prementum–palpi articulation joints (J1, J2) to capture the prey (Figure 2B). Such joints are also observed in damselfly larvae mouthparts and they were qualitatively observed to be sclerotized using a confocal laser scanning method [16]. These joints undergo repeated stress cycles during opening and closure of the palpi during the lifetime of the nymph and are possibly resistant to wear.

Setae were observed along the rims of the two palpi (Figure 2A). These were hypothesized to be chitin-based microstructures that help in detecting either chemical or mechanical signals [17]. Sensillae were observed in a row along the line of contact of the two identical palpi (Figure 2C). We also observed other types of sensillae that were uniformly distributed on the inner surface of palpi (Figure 2D). The sensillae along the edges where the palpi come in contact were observed to be in groups of two and three, possibly performing a mechanical function (Figure 2E). They appeared to be very similar to the trichoform sensillum in insects. Such sensillae were observed on the labium of *Crocothemis servilia* [18]. In the antennae of the broad-bodied chaser (*Libellula depressa*), a coeloconic sensillum was observed and its function is possibly to detect temperature variations [19]. We also observed smaller sized sensillae inside sockets (Figure 2E), which appeared to be similar to type ii coeloconic sensillae [20]. The overall macrostructural and microstructural details suggested that the prey-capturing mechanism is a complex grasping mechanism with various sensillae that might aid in detecting the surroundings.

### 3.2. Labial Movement during Prey Capture

Our high-speed videos allowed us to view the detailed protraction and retraction of labium during the prey capture (Appendix A). The time taken for full labial extension during the capture of the prey was 63 ± 24 ms (Table 1). This was in close agreement with the time of protraction reported for the Canada darner (*Aeshna Canadensis*), i.e., ~80 ms [11]. The fastest known animal strike by the peacock mantis shrimp (*Odontodactylus scyllarus*) was observed to occur in ~3 ms. The dragonfly nymph predatory strike is not as fast as that of the mantis shrimp, but it includes an additional task of opening and closing the labial palpi during the time of the strike. The total time taken for the protraction and retraction of labium during the capture of the prey was 187 ± 54 ms (Table 1). The total time of capture and sweeping, in insects such as the praying mantis (*Tenodera aridifolia sinenis*), was found to be ~70 ms [21]. The times taken by individual specimens are listed below for more details (Table 1). The protraction of the prey-capturing labium, in co-ordination with palpi, involves a kinetic movement of muscles controlled by joints (Figure 3 and Figure 4). During the retraction of the labial palpi that encase the prey, we observed a jet expulsion from the anal orifice. This supports the earlier claim that the labium protraction and retraction is also assisted by a hydraulic mechanism.

The drag forces acting on the labial palpi during protraction can be a combination of viscous drag due to fluid friction and an ‘‘added mass’’ force. The force due to an ‘‘added mass’’ effect can be understood by noting that the water surrounding the labial palpi is relatively stationary before prey capture, and once the labial palpi open out, it must go against the inertial resistance of surrounding water (between panels B and C in Figure 4) for a short duration. This unsteady force can be a significant contributor to the overall drag experienced by the labial palpi. For example, in a study based on swimming scallops, it was shown that the added mass corresponding to the transient opening of the shells results in large hydrodynamic forces [22].

We observe that the labial palpi open first with the aid of rotational freedom in the vertical plane and the labium then extends further towards the prey. This is in agreement with earlier observations made on the predatory strike of the *Aeschna nigroflav* nymph [10]. During this period, when the labium is protracting along with the palpi opening out, viscous drag can be expected to be the dominant force compared to added mass effects, primarily because the labial palpi now slice forward rather than pushing water in the lateral direction. We observe that the points (Figure 5: L1, R1) that correspond to the triangular edges of the labial palpi are rotated vertically upwards to achieve an optimal angle of attack and thereby reduce the viscous drag with the help of each labial palpi curvature. During this process, the edge of the ligula of labium is pushed forward and does not undergo any rotation (Figure 5: B1).

Thus, the dragonfly nymph appears to move the labial palpi in such a way to reduce the viscous drag during prey capture and thereby reduce the energy spent. The prementum–palpi articulation joints (J1, J2), which appear to be sclerotized because of the darker coloration as shown in Figure 5, are in agreement with the observations made in a different species of damselfly nymph [16]. Such joint mechanisms with a sclerotized cuticle embedded in a flexible cuticle were also observed in the leg joints of frog hoppers [23]. They play an important role in facilitating rotation of the labial palpi during multiple openings and closures during the lifetime of the nymph.

### 3.3. Mechanical Properties of the Mandibles

Force-depth curves from the microindentation data of mandibles that enable the cutting of captured food show good repeatability (Figure 6A,B). The data analysis shows that the elastic modulus and hardness of the mandibles are 9.1 ± 1.9 GPa and 0.85 ± 0.13 GPa. The mandible cross-section shows different regions of sclerotized and non-sclerotized regions, and that the mandible is a hollow structure (Figure 6C). Such hollow mandibles were shown to be optimal structures with higher stiffness than the corresponding solid structures with the same material volume without compromising the functionality [24]. The indentations from different locations of the mandibles show the Vickers indenter impression (Figure 6D,E). The measured mechanical properties are similar to that of termite mandibles (Hardness: 0.4–1.2 GPa and Young’s modulus 6–11 GPa) of various species [25] and hardness (0.7–0.9 GPa) of adult leaf cutting ant mandibles [26]. Thus, the mandibles of the nymph appear to be stiff and hard enough to bite through the flesh of the prey.

Our nanoindentation results show that there is a spread of the data from one region to another, demonstrating the regional differences in mechanical properties (Figure 7 and Figure 8). The measurements in the region away from the tip (Figure 7) show less variation than the measurements in the tip region (Figure 8). The elastic modulus varies from 6.6 to 8.3 GPa in the base region and from 2 GPa to 5.5 GPa near the tip region, whereas the hardness varies in the base location from 0.34 GPa to 4.4 GPa (Figure 7D) and in the tip location from 0.4 GPa to 0.56 GPa (Figure 8D). The higher hardness tip region of the mandibles is known to reduce the wear, as observed in the jaws of underwater organisms such as *Nereis virens* [27]. Most notably, a demarcation in the properties in the mandible tip region is observed in the Young’s modulus and hardness maps, suggesting a gradation of mechanical properties in the mandibles (Figure 8A,B). Such gradients were qualitatively observed in the damselfly (*Erythromma najas*) larval mandibles [16]. These gradients can be attributed to the difference in microstructure or metal mediated stiffening resulting in higher mechanical properties [27,28,29]. Simulations carried out on mandibles with gradation in the mechanical properties have been shown to help in reducing stress concentrations and also to increase damage resilience [30]. The data also suggests that the properties are dependent on the size of the indenter, indicating the role of scaling effects on the mechanical properties.

## 4. Conclusions

In this study, we discuss the biomechanical aspects involved in the prey-capturing mechanism of the dragonfly nymph. The novel aspects of this study include a demonstration of labial palpi slicing movement underwater to grab the prey, which suggests that it is a drag reduction approach, and the first report of dragonfly nymph mandible properties. We observed that the mechanism deploys in a fraction of a second in spite of the inertial and drag forces that are encountered underwater. We also observed a pressure release from the abdomen of the nymph during the retraction of the palpi, supporting the earlier hypothesis of usage of hydraulic mechanism. The nymph mandibles were observed to have high hardness similar to that of other insect mandibles to reduce wear and to also have gradation in mechanical properties to mitigate the stress concentrations.

The present observations on the movement and shape of the labial palpi, in conjunction with its sensory capabilities, provide an opportunity towards the design of bioinspired underwater manipulators. This can indeed support the growing interest in designing robotic manipulators that are soft, as compared to the more conventional rigid mechanisms [31]. Taking inspiration, a soft and foldable manipulator can be designed to aid in live specimen collection in deep waters without damaging them. Specifically, by mimicking the prey capturing mechanism, an apparatus can be designed in such a way that it minimizes the hydrodynamic drag, thereby making them more energy efficient. Additionally, taking inspiration from the labium, soft actuators for rapid movements can be designed by embedding stiff elements inside the soft materials. We also observed that the mandibles of the nymphs have gradation in material properties and these principles can be used to design micro-cutting tools in special applications. Furthermore, the studies aimed at understanding the chemical components and orientation of microstructural elements in the mandibles may aid in development of graded materials with specifically tuned properties for the underwater-use materials. Future studies aimed at examining the joint morphology could improve the understanding of its role in the capturing mechanism. Additionally, the present study opens up questions on how the dragon fly nymphs use their prey-capturing mechanism in the presence of other predators, and this is also a possible future direction for this work.

## Figures and Tables

**Figure 1 materials-14-00559-f001:**
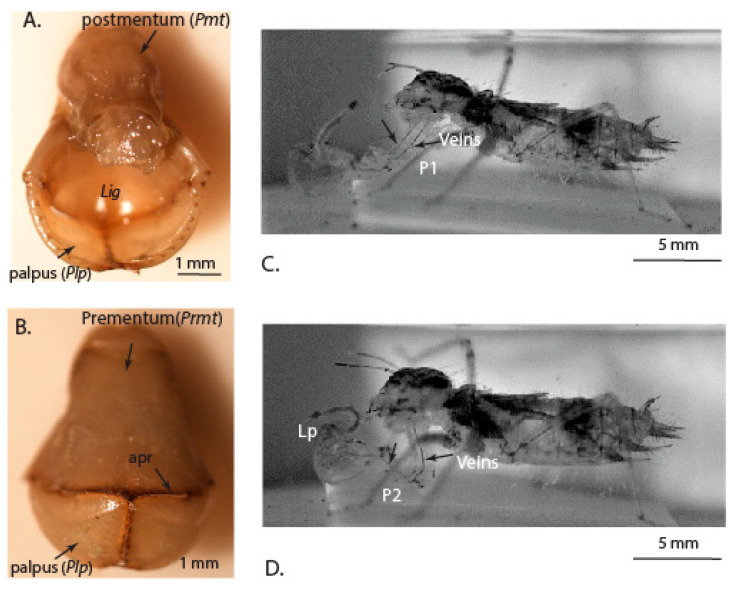
Images showing the labium. (**A**) Dorsal view with Ligula and palpi; (**B**) Ventral view with palpi and *apr*, apico-premental ridge; and (**C**) view of protracted labium and (**D**) view of the labium during retraction after the prey is captured.

**Figure 2 materials-14-00559-f002:**
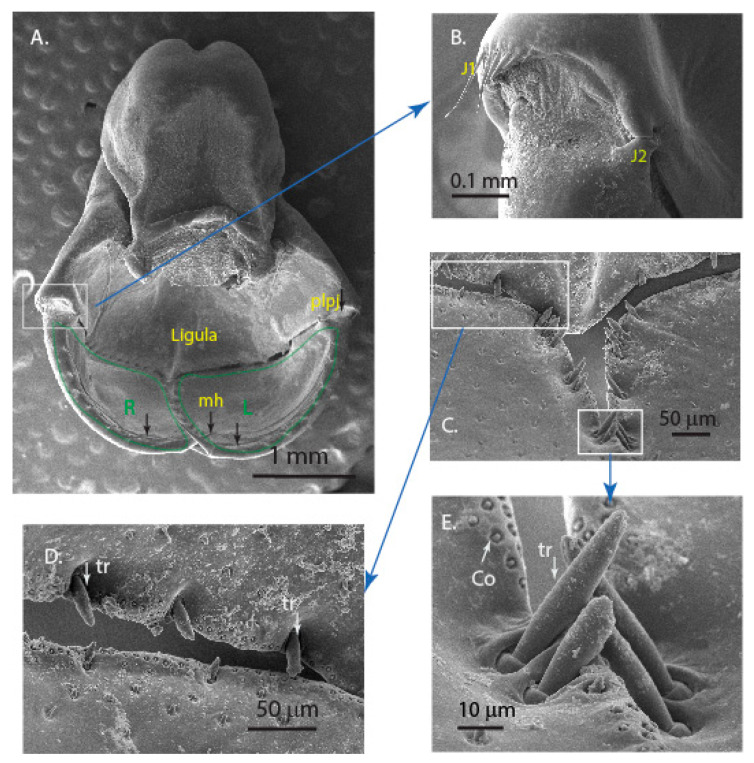
Scanning Electron Micrographs of (**A**) the labium and palpi (L and R) with setae (black arrows); (**B**) the prementum-labial palpus joint (*plpj*) and the articulation joints (J1, J2), which play a crucial role in the movement of the labial palpi during the prey capture; (**C**) sensillae observed on the edges and surface of the labial palpi; (**D**) junction of the three mouthparts showing the various types of sensillae; and (**E**) zoomed region showing the sensillae in pairs along the line of contact of two half labial palpi.

**Figure 3 materials-14-00559-f003:**
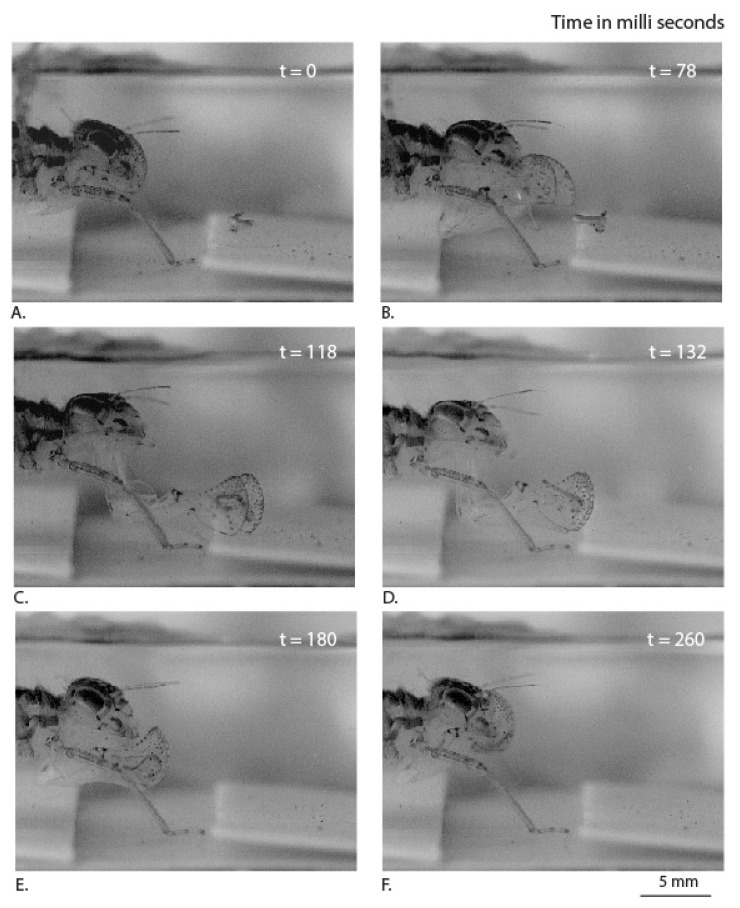
Sequence of movements (side view) used by the dragonfly nymph labium steps to capture mosquito larvae, at different time points (**A**–**F**).

**Figure 4 materials-14-00559-f004:**
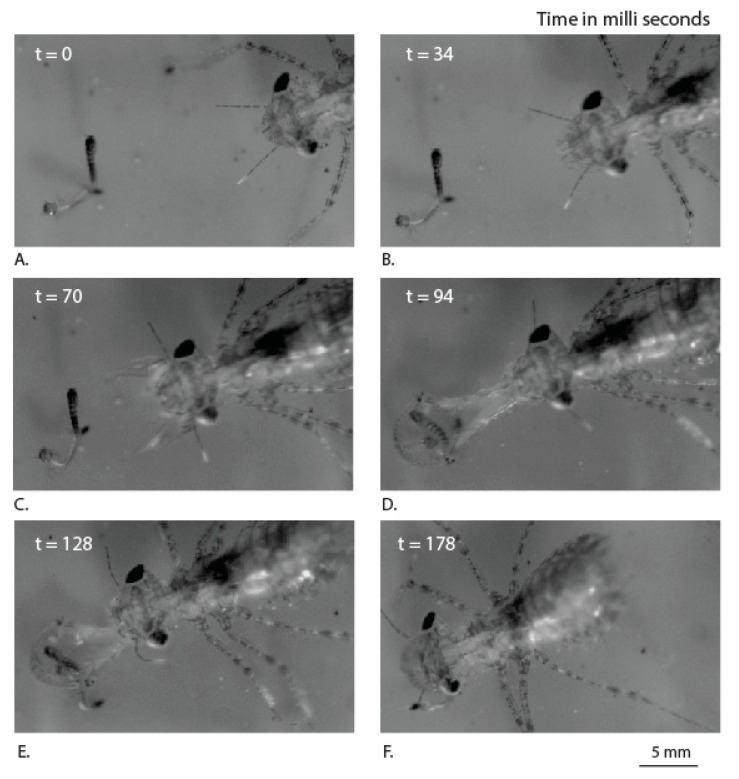
Sequence of movements (top view) used by the dragonfly nymph labium to capture the mosquito larva, at different time points (**A**–**F**).

**Figure 5 materials-14-00559-f005:**
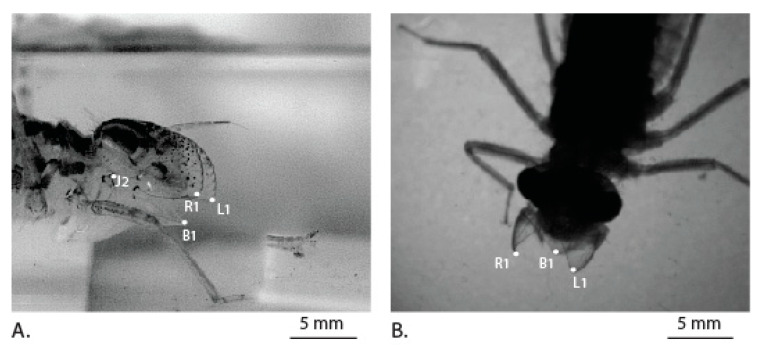
Position of palpi just before protraction of the labium. (**A**) Side view. (**B**) Top view.

**Figure 6 materials-14-00559-f006:**
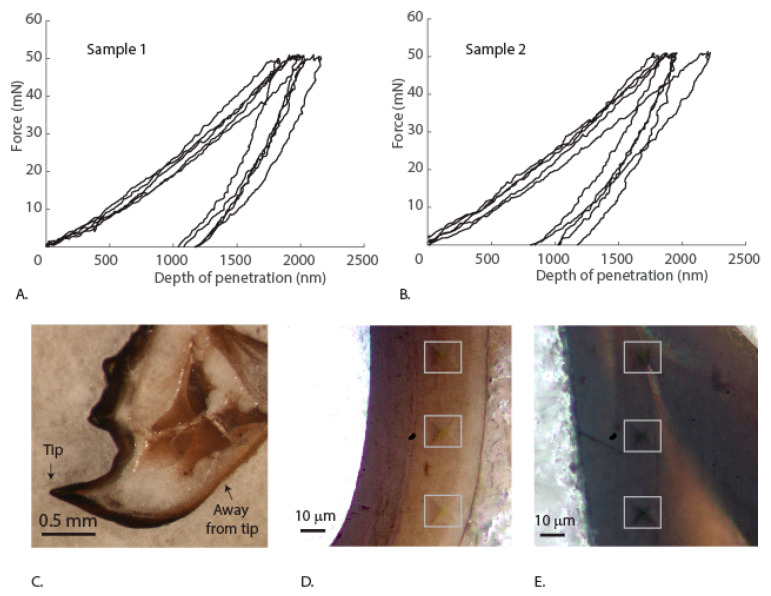
(**A**,**B**) Force depth curves of microindentation from sample 1 and sample 2. (**C**) Polished cross-section of the mandible. (**D**) Vickers indentation impressions in the “away from tip” region. (**E**) Vickers indentation impressions in the “away from tip” region.

**Figure 7 materials-14-00559-f007:**
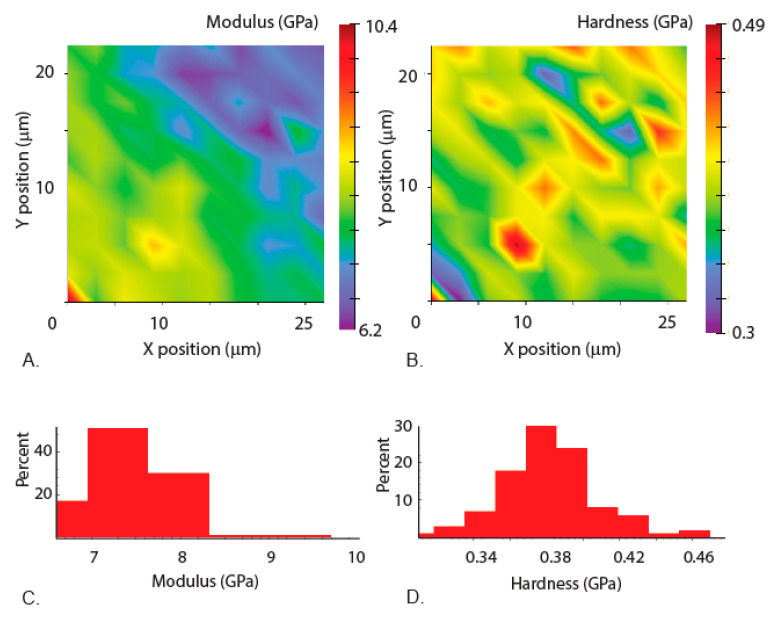
Measurements from the region away from the tip (as shown in Figure 6C). (**A**) Elastic modulus mapping across the layers. (**B**) Hardness mapping across the layers. (**C**) Corresponding elastic modulus values sorted into bins to show the variation and percentage. (**D**) Corresponding hardness values sorted into bins to show the variation and percentages.

**Figure 8 materials-14-00559-f008:**
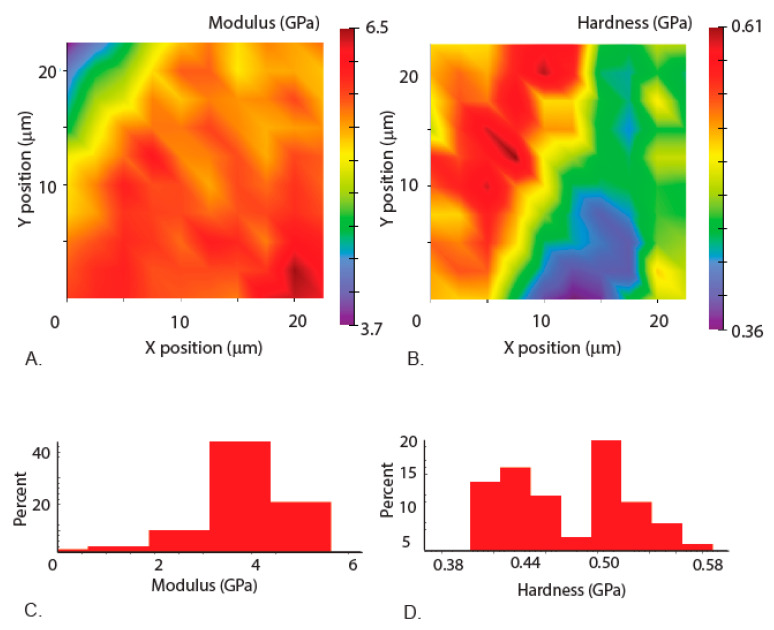
Nanoindentation measurements from the tip region (as shown in Figure 6C). (**A**) Elastic modulus mapping across the layers. (**B**) Hardness mapping across the layers. (**C**) Corresponding elastic modulus values sorted into bins to show the variation and percentage. (**D**) Corresponding hardness values sorted into bins to show the variation and percentages.

**Table 1 materials-14-00559-t001:** Time taken for protraction of labial palp and capture of the prey.

Sample	Time for Protraction (ms)	Total Time for Capture (ms)
1	60	225
2	36	114
3	40	124
4	78	210
5	64	240
6	100	208
Average	63 ± 24	187 ± 54

## Data Availability

Data is contained within the article or Appendix A.

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
