# Peer review of "Prey Capturing Dynamics and Nanomechanically Graded Cutting Apparatus of Dragonfly Nymph"

_materials, 2021, doi:10.3390/ma14030559_

Round 1

Reviewer 1 Report

This article is very interesting in terms of the wonder of the nature, and bioinspiration & biomimetics. This research described an unique engineering approach to reveal the sophisticated biomechanism using Hi speed video, SEM and nanoindentation test. In order to enhance the value of this article, please consider following review comments.

  1. In Introduction, the author should firstly explain the unique method of the prey capturing, before describing the mouth parts and their function, which will be a great help to understand this article with clear interpretation of the actual underwater images.
  2. In the caption of Fig.7 and Fig.8, it is recommended to describe that the measured regions have been indicated in Fig.6.
  3. In Discussion, it is preferable to suggest prospects of this research, and the possible application fields of the obtained knowledge, e.g. hydrodynamics and robotics for undersea operations and the sea floor developments. As an article for the Journal of Material, the author should describe technological significance of this study.

Author Response

This article is very interesting in terms of the wonder of the nature, and bioinspiration & biomimetics. This research described a unique engineering approach to reveal the sophisticated biomechanism using Hi speed video, SEM and nanoindentation test. In order to enhance the value of this article, please consider following review comments.

We thank the reviewer for their comments on the manuscript that helped us improve it. We present below a point-by-point response to the comments including the changes that have been made to the manuscript. All the comments were addressed below (in blue) and corresponding changes in the article are thus implemented (in violet).

In Introduction, the author should firstly explain the unique method of the prey capturing, before describing the mouth parts and their function, which will be a great help to understand this article with clear interpretation of the actual underwater images.

We thank the reviewer for the suggestion, and have now included the following text to explain the method of capture in the Introduction section.

Page 3: line 66-78

“Prey capturing by a dragonfly nymph involves the steps of visually locking on the prey to be captured and also the pressure build-up in the abdomen with water that helps in generating quick forward movement using propulsion, in certain cases. The predatory strike in dragonfly nymphs is performed by using the labium that is modified into a prehensile mask for capturing the fast moving preys (Büsse et al., 2017). This involves gradual opening up of the two cup-shaped labial palpi and moving them forward with the help of unfolding prementum and postmentum parts, similar to extension of human arms with the elbow and shoulder joints. Depending on the distance of the prey from the mouth part, the forward movement was observed to be assisted by the movement of the body. Once the palpi reach around the prey, they are brought together to form a closed cup shape to grab the prey.  Once the prey is captured, it is drawn towards the mouth to feed on it.”

In the caption of Fig.7 and Fig.8, it is recommended to describe that the measured regions have been indicated in Fig.6.

We have added the following text in the both figure captions as suggested.

“(as shown in Fig 6C)”

In Discussion, it is preferable to suggest prospects of this research, and the possible application fields of the obtained knowledge, e.g. hydrodynamics and robotics for undersea operations and the sea floor developments. As an article for the Journal of Material, the author should describe technological significance of this study.

We have added the following text to suggest some possible applications and also the technological significance of the study.

Page 18: line 295 - 310

“The present observations on the movement and shape of the labial palpi, in conjunction with its sensory capabilities, provide an opportunity towards the design of bioinspired underwater manipulators. This can indeed support the growing interest in designing robotic manipulators that are soft as compared to the more conventional rigid mechanisms (Rus and Tolley, 2015). Taking inspiration, a soft and foldable manipulator can be designed to aid in live specimen collection in deep waters without damaging them. Specifically, by mimicking the prey capturing mechanism, an apparatus can be designed in such a way that it minimizes the hydrodynamic drag, thereby making them more energy efficient. Also, taking inspiration from the labium, soft actuators for rapid movements can be designed by embedding stiff elements inside the soft materials. We also observed that the mandibles of the nymphs have gradation in material properties and these principles be used to design micro-cutting tools in special applications. Future studies aimed examining the joint morphology can improve the understanding of its role in the capturing mechanism. Also, the studies aimed at understanding the chemical components and orientation of microstructural elements in the mandibles may aid in development of graded materials with specifically tuned properties for the underwater-use materials.”

This article is very interesting in terms of the wonder of the nature, and bioinspiration & biomimetics. This research described a unique engineering approach to reveal the sophisticated biomechanism using Hi speed video, SEM and nanoindentation test. In order to enhance the value of this article, please consider following review comments.

We thank the reviewer for their comments on the manuscript that helped us improve it. We present below a point-by-point response to the comments including the changes that have been made to the manuscript. All the comments were addressed below (in blue) and corresponding changes in the article are thus implemented (in violet).

In Introduction, the author should firstly explain the unique method of the prey capturing, before describing the mouth parts and their function, which will be a great help to understand this article with clear interpretation of the actual underwater images.

We thank the reviewer for the suggestion, and have now included the following text to explain the method of capture in the Introduction section.

Page 3: line 66-78

“Prey capturing by a dragonfly nymph involves the steps of visually locking on the prey to be captured and also the pressure build-up in the abdomen with water that helps in generating quick forward movement using propulsion, in certain cases. The predatory strike in dragonfly nymphs is performed by using the labium that is modified into a prehensile mask for capturing the fast moving preys (Büsse et al., 2017). This involves gradual opening up of the two cup-shaped labial palpi and moving them forward with the help of unfolding prementum and postmentum parts, similar to extension of human arms with the elbow and shoulder joints. Depending on the distance of the prey from the mouth part, the forward movement was observed to be assisted by the movement of the body. Once the palpi reach around the prey, they are brought together to form a closed cup shape to grab the prey.  Once the prey is captured, it is drawn towards the mouth to feed on it.”

In the caption of Fig.7 and Fig.8, it is recommended to describe that the measured regions have been indicated in Fig.6.

We have added the following text in the both figure captions as suggested.

“(as shown in Fig 6C)”

In Discussion, it is preferable to suggest prospects of this research, and the possible application fields of the obtained knowledge, e.g. hydrodynamics and robotics for undersea operations and the sea floor developments. As an article for the Journal of Material, the author should describe technological significance of this study.

We have added the following text to suggest some possible applications and also the technological significance of the study.

Page 18: line 295 - 310

“The present observations on the movement and shape of the labial palpi, in conjunction with its sensory capabilities, provide an opportunity towards the design of bioinspired underwater manipulators. This can indeed support the growing interest in designing robotic manipulators that are soft as compared to the more conventional rigid mechanisms (Rus and Tolley, 2015). Taking inspiration, a soft and foldable manipulator can be designed to aid in live specimen collection in deep waters without damaging them. Specifically, by mimicking the prey capturing mechanism, an apparatus can be designed in such a way that it minimizes the hydrodynamic drag, thereby making them more energy efficient. Also, taking inspiration from the labium, soft actuators for rapid movements can be designed by embedding stiff elements inside the soft materials. We also observed that the mandibles of the nymphs have gradation in material properties and these principles be used to design micro-cutting tools in special applications. Future studies aimed examining the joint morphology can improve the understanding of its role in the capturing mechanism. Also, the studies aimed at understanding the chemical components and orientation of microstructural elements in the mandibles may aid in development of graded materials with specifically tuned properties for the underwater-use materials.”

Reviewer 2 Report

This manuscript studied the biomechanical aspects of prey capture. However, the author needs to explain more about the materials probably developed from these mechanics; the following are minor comments

  1. The manuscript looks more biological study, authors need to discus more on materials that can be developed.
  2. If possible author can zoom Co structure in Fig. 2.
  3. Authors need to give details of the nymphs of globe skimmers studied in section 2.1. at what stage.
  4. What is the capturing mechanism need to explain in detail.
  5. What is the possible mechanics in presence of the prey competiters.

Reviewer 3 Report

In the presented manuscript, Pugno  et al. described Prey capturing and cutting apparatus of dragonfly nymph. Presented article is properly designed and well presented. However, I do not see the potential application of the research described in the paper.

The Authors should focus on future prospects and indicate novelty of their studies. It is only stated that presented results should be useful in bioinspired underwater deployable mechanisms. Also, References section need to be refreshed and implemented.

Round 2

Reviewer 2 Report

The authors defended all the comments nicely. The manuscript can be accepted now.

Reviewer 3 Report

The Authors did a good job and presented manuscript can be further processed.